# Perceptions on the Implementation of a School Nursing Pilot Programme in the Canary Islands

**DOI:** 10.3390/nursrep15020048

**Published:** 2025-01-31

**Authors:** Aixa Fernández-Hernández, Laura Figueroa-Martín, Sandra-Jesús González-Betancor, Begoña Reyero-Ortega, Héctor González-de la Torre, Claudio-Alberto Rodríguez-Suárez

**Affiliations:** 1Los Realejos Primary Care Facility, Tenerife Primary Care Management Board, The Canary Islands Health Service (SCS), 38410 Santa Cruz de Tenerife, Canary Islands, Spain; aferherr@gobiernodecanarias.org; 2Güimar Primary Care Facility, Tenerife Primary Care Management Board, The Canary Islands Health Service (SCS), 38500 Santa Cruz de Tenerife, Canary Islands, Spain; lfigmar@gobiernodecanarias.org; 3Arucas Primary Care Facility, Gran Canaria Primary Care Management Board, The Canary Islands Health Service (SCS), 35400 Las Palmas, Canary Islands, Spain; sgonbet@gobiernodecanarias.org; 4Support Unit to the Directorate of the Canary Islands Health Service, The Canary Islands Health Service (SCS), 38005 Santa Cruz de Tenerife, Canary Islands, Spain; breyort@gobiernodecanarias.org; 5Nursing Department, Faculty of Health Sciences, University of Las Palmas de Gran Canaria (ULPGC), 35016 Las Palmas, Canary Islands, Spain; 6Research Support Unit, Insular Maternal and Child University Hospital Complex (CHUIMI), The Canary Islands Health Service (SCS), 35016 Las Palmas, Canary Islands, Spain

**Keywords:** school nursing, nursing, nursing care, nurse’s role, qualitative research

## Abstract

**Background/Objectives**: School nursing is a nursing practice focused on promoting child health within the school and community environment, integrating healthcare with the educational process. The aim was to explore the experiences of nurses and teachers regarding the implementation of a school nursing pilot programme in primary education in the Canary Islands (Spain). **Methods**: A phenomenological study was conducted during the 2023/2024 academic year. In-depth interviews were conducted between May and June 2024 until data saturation was achieved. The interviews were transcribed and analysed using descriptive and interpretative thematic analysis. Intentional and co-occurrence coding methods were employed, followed by triangulation using Atlas-Ti software (version 24). Ethical approval was obtained (code: 2023-216-1). **Results**: A total of 21 informants (7 nurses and 14 teachers) were interviewed. Nurses revealed eight subthemes grouped into two main themes: Role of School Nurses (workload, school nurses’ experiences, career opportunities, and the school nurse profile) and School Nursing Project (suggested improvements, identified weaknesses, time management, and improvement needs). Regarding teachers, six subthemes were identified, also grouped into two main themes: Role of School Nurses (approach to school nursing and the importance of the presence of school nurses) and School Nursing Project (expectations, improving children’s health, school health, and experiences from the pilot programme). **Conclusions**: The pilot programme has been well received, showing a positive impact on students’ health. While successfully integrating nurses into schools, improvements are needed in working conditions, resource allocation, and specialized training to enhance its effectiveness and sustainability. Strengthening inter-professional collaboration between healthcare and education sectors and involving teachers in planning health activities are essential. Expanding the coverage and ensuring a consistent presence of school nurses would further build trust, improve chronic health management, and promote healthy habits from an early age. This programme illustrates the potential of nurses to transform schools into spaces for comprehensive health promotion.

## 1. Introduction

The National Association of School Nurses defines school nursing as a specialised nursing practice focused on protecting and promoting the health of children within the school and community environment. This role facilitates their optimal development and academic success through ethical, evidence-based practice [1]. School nursing bridges healthcare and education [1,2], leveraging the school setting as a conducive space for health promotion and the encouragement of healthy behaviours [2].

The role of school nurses in education has been widely studied, particularly concerning the management of prevalent chronic conditions [3] such as diabetes [4], childhood obesity [5], respiratory diseases [6], mental health [7], and the administration of pharmacological treatments [8]. Their involvement in health promotion campaigns [2] has also been examined, including education on menstruation [9], healthy sleep habits [10], child abuse prevention [11], heatstroke prevention [12], and school vaccination programmes [13], among others. Moreover, their presence and impact during the COVID-19 pandemic [1,14] have been the subject of significant research. Other explored topics include school nurses’ interactions with families [15], the impact of school nursing programmes on student health [16,17], and the training of teachers for emergency response [18]. However, there remains a need to generate evidence demonstrating school nurses’ effectiveness to policymakers and to expand research on their skills and competencies in diverse international contexts [2].

Despite the growing interest in this field, the study of professionals’ perceptions and experiences regarding the school nurse role has been less extensively covered in the scientific literature [13,15], underscoring the importance of exploring this aspect through qualitative approaches.

Against this background, the Regional Government of the Canary Islands has developed and implemented a school nursing pilot programme in the Autonomous Community. As an insular territory, the programme’s implementation posed organisational challenges requiring close collaboration among institutions of varied natures and characteristics, as well as among professionals with diverse academic and professional backgrounds. This study aims to explore the experiences of nurses and teaching staff regarding the implementation of the school nursing pilot programme in early years and primary education in the Canary Islands.

## 2. Materials and Methods

### 2.1. Design

A qualitative study with a phenomenological approach was conducted, following van Manen’s methodological proposal [19,20], which integrates Husserl’s descriptive perspective with Heidegger’s interpretative hermeneutics [21]. Van Manen’s approach provides a robust framework for exploring and understanding how participants perceive and assign meaning to their experiences in a specific context. Its emphasis on analysing essential meaning, critical reflection, and language as an analytical tool strengthens methodological coherence and alignment with the study’s aims.

### 2.2. Study Setting and Recruitment

The Canary Islands, a Spanish archipelago in the Atlantic Ocean off the northwest coast of Africa, consist of seven main islands with rich cultural diversity and a distinctive educational context shaped by their insular nature. Most schools are located on larger islands such as Gran Canaria and Tenerife, while smaller islands such as El Hierro and La Gomera have fewer schools due to lower population densities. The population is primarily Canarian of Spanish origin, with a significant immigrant community from Hispano-America, Africa, and Eastern Europe. This diversity enriches school environments but also presents challenges related to linguistic and cultural inclusion. Schools promote healthy habits, although childhood obesity rates are higher than the national average. This unique and diverse setting requires tailored educational strategies to address cultural and regional disparities effectively. The study population consisted of two groups of participants. The first group included nurses involved in the pilot programme (*n* = 20) for the implementation of the school nurse role in primary education institutions in the Canary Islands, Spain. The second group comprised teachers from 60 educational centres participating in the pilot programme. Since the participants were either nurses in the program or teachers at schools included in the program, they were already aware of the study’s objectives. The initial selection of participants was conducted using convenience sampling, aiming to interview nurses and teachers from each island, as socio-economic contexts and school resources vary significantly due to their insular nature. This was followed by theoretical sampling to recruit key informants, thereby maximising the richness and diversity of discourse surrounding the studied phenomenon until data saturation was reached. To address potential bias, efforts were made to identify individuals within the target population who held critical perspectives on the school nursing project. No participants refused to participate or dropped out.

### 2.3. Inclusion and Exclusion Criteria

Inclusion criteria encompassed nurses participating in the school nursing programme and teachers from the educational centres involved in the programme during the 2023/2024 academic year. Teachers were included regardless of whether they participated in health promotion and disease prevention activities. No exclusion criteria were applied.

### 2.4. Data Collection

Data were collected through a sole in-depth interview (AFH, SJGB, LFM) conducted both face-to-face and online [22]. In-person interviews were held in offices within the educational institutions, ensuring a comfortable, interruption-free environment. For virtual interviews, the Webex^®^ (San Jose, CA, USA) platform (the official teleconferencing tool of the Regional Government of the Canary Islands) was used, guaranteeing a stable, seamless connection. Interviews occurred between May and June 2024, lasting between 60 and 120 min, with prior scheduling coordinated with participants. No one else was present during the interviews. Data collection concluded once data saturation was reached.

The interview script was developed using Atlas-Ti^®^ software (version 24.1.0), leveraging its artificial intelligence (AI) module to generate context- and objective-driven questions. These were subsequently refined to align with the intended analytical categories established during coding. Questions in the semi-structured script were tailored to each participant group, as detailed in Table 1.

Sociodemographic variables were collected, including island of residence, age, sex, and years of professional experience. For nurses, additional information was gathered regarding their main primary care facility, postgraduate training, and the number of assigned educational institutions. For teachers, variables included class tutoring, teaching level, and subjects, as well as the number of students at their school.

### 2.5. Data Analysis

The face-to-face interviews were audio-recorded, while the online interviews were audio–video-recorded. Both were transcribed verbatim to be incorporated into the thematic analysis units, along with researchers’ field notes. To perform the transcriptions, the Good-Tape software (https://goodtape.io/) was used. The analysis followed an intentional coding strategy [23,24], supported by Atlas Ti software (version 24.1.0) and based on Strauss’ methodology, which consists of two phases. Firstly, a phenomenological reduction was conducted by two researchers (H.G.-d.l.T., C.-A.R.-S.), identifying descriptive units of meaning (UM) and grouping them into subthemes. Then, a phenomenological interpretation was applied through inductive data analysis to understand the studied phenomenon. Finally, selective coding was performed by merging similar subthemes to identify central theoretical categories or overarching themes that explain the phenomenon [25]. Additionally, a comparative axial coding of the data was conducted through co-occurrence analysis of units of meaning and subthemes. The analysis of co-occurrences involves identifying and examining the relationships between categories that frequently appear together within. This method enables a deeper exploration of how concepts are interconnected within participants’ narratives, uncovering patterns, associations, and emerging themes. By highlighting these interrelations, co-occurrence analysis provides valuable insights into the structure and meaning of the phenomena, enhancing the overall rigor and depth of qualitative analysis [26]. Discrepancies in the analysis were discussed and resolved through triangulation among the research team members until a consensus was reached.

### 2.6. Rigour and Reflexivity

To ensure rigour, the quality criteria proposed by Lincoln and Guba [27,28] were followed. Credibility was achieved through detailed data collection at all stages, with descriptive processes verified by informants during the review of results. The coding process involved multiple researchers independently coding the data, followed by regular discussions to reach consensus on the identified subthemes and themes, ensuring consistency in the interpretation. Transferability was ensured through a comprehensive description of the setting, participants, context, and methods used. Dependability was evaluated via an external review of the results by an expert unfamiliar with the data collection and analysis. Confirmability was established through researcher and data triangulation between the transcripts and the researcher’s field notes, wherein all team members read the transcripts and reached a consensus on UM, subthemes, and themes. The research team also engaged in continuous reflexivity to address potential biases and influences. Two of the researchers (HGdlT, CARS) were men with PhD degrees and prior experience in qualitative research. The remaining researchers (AFH, SJGB, LFM, BRO) were women, nurses without prior experience in qualitative research, who were employed as school nurses during the study. These researchers, who conducted the interviews, were school nurses with a strong understanding of both the context and the study’s focus. They underwent prior training to ensure their expertise and preparedness contributed to the rigor and depth of the data collection process. However, the nurses had no previous relationship with the teachers involved in the study. The manuscript was prepared in accordance with the Consolidated Criteria for Reporting Qualitative Research (COREQ) guidelines [29].

### 2.7. Ethical Considerations

Although the design of this study did not anticipate potential harm to participants, ethical approval was sought and obtained from the Ethics Committee for Research with medicinal products (ECRmp) for the Las Palmas—Dr. Negrín University Hospital in Gran Canaria (code: 2023-216-1). Additionally, the University Hospital Complex of the Canary Islands in Tenerife reviewed the study and determined that no further evaluation was required. Participant anonymity and voluntary participation were ensured through the use of pseudonyms, and data were handled in compliance with the Spanish Organic Law 3/2018 of 5 December, on Personal Data Protection and Guarantee of Digital Rights. Authorisation to record interviews was requested following the signing of informed consent by participants. All recordings and transcripts were securely stored in accordance with current legislation, with access restricted exclusively to the research team.

## 3. Results

### 3.1. Sociodemographic Results

A total of *n* = 21 informants were interviewed. Of these, *n* = 7 were nurses, each assigned to between two and four schools. The sociodemographic characteristics of this group are detailed in Table 2.

Regarding the teachers, *n* = 14 participants from various primary education centres were included. Their characteristics are detailed in Table 3.

### 3.2. Descriptive Results

In the interviews conducted with nurses, a total of *n* = 221 verbatims were coded and grouped into *n* = 116 UM, which were organised into *n* = 8 subthemes and *n* = 2 main themes (Role of School Nurses and School Nursing Project). The theme Role of School Nurses included the following subthemes: Workload (*n* = 19 UM), School Nurses’ Experiences (*n* = 64 UM), Career Opportunities (*n* = 82 UM), and The School Nurse Profile (*n* = 136 UM). The theme School Nursing Project encompassed the following subthemes: Suggested Improvements (*n* = 28 UM), Identified Weaknesses (*n* = 16 UM), Time Management (*n* = 80 UM), and Improvement Needs (*n* = 37 UM).

Regarding the interviews conducted with teachers, a total of *n* = 103 verbatims were coded and structured into *n* = 31 UM. These units were grouped into *n* = 6 subthemes, distributed across *n* = 2 main themes (Role of School Nurses and School Nursing Project). The theme Role of School Nurses included the following subthemes: Approach to School Nursing (*n* = 19 UM) and Importance of the Presence of School Nurses (*n* = 27 UM). The theme School Nursing Project encompassed the following subthemes: Expectations (*n* = 25 UM), Improving Children’s Health (*n* = 48 UM), School Health (*n* = 53 UM), and Experiences from the Pilot Programme (*n* = 43 UM). The complete list of themes, subthemes, and UM is detailed in Table 4.

The co-occurrences among subthemes identified in the interviews with nurses revealed significant discursive relationships. Notable connections included the co-occurrence between Time Management and The School Nurse Profile (*n* = 80), as well as between Experiences of School Nurses and Career Opportunities (*n* = 52). For the teachers, the most relevant co-occurrences were observed between Experiences of the Pilot Programme and Improving Children’s Health (*n* = 36). Other notable co-occurrences included those between Importance of the Presence of School Nurses and School Health (*n* = 27), as well as between School Health and Expectations (*n* = 23). The full list of co-occurrences is available in Appendix A.

#### 3.2.1. Theme: Role of School Nurses

The theme Role of School Nurses was addressed by both nurse and teacher participants. For the nurses, the most prominent subthemes included:The School Nurse Profile

This subtheme was most significant in the nurses’ accounts, emphasising how school nursing provides career opportunities to expand their skills and scope. Participants shared their often-divergent expectations, highlighting the motivational and educational aspects present since the project’s inception. They stressed the importance of selecting professionals whose profiles align with the school environment:

“You’ve really got to enjoy working with kids … Paediatric or community nursing specialization, or even a university-level course, would be a good idea … Plus, training in basic and advanced paediatric life support is crucial, just in case something happens. And you need to be confident when it comes to teaching …”(N6)

“Paediatric training is essential, not only for promoting health but also for clinical issues. We’re the ones on-site, and if something happens, we’ll be the ones they call. Tech skills and a knack for research should also be must-haves for a school nurse”(N4)

Selecting motivated professionals was another key element for the success of the project:

“At the beginning, it was awful … But as we completed training and prepared workshops, things became more motivating. It felt like I was making a difference, and people come to me. It was a whole different feeling”(N2)

School Nurses’ Experiences and Career Opportunities

The co-occurrences between School Nurses’ Experiences and Career Opportunities highlighted their enthusiasm for working in school:

“The children know you’re someone they can turn to at any time—not just the children, but the teachers too”(N5)

However, these experiences were influenced by disparities in the selection processes:

“In my case, I was offered the position before it was even officially announced …”(N1)

“I didn’t have to go through interviews. I called management, and they said, ‘Yes, you’re the one’. The rest was for applicants—time worked in primary care and seniority”(N2)

This highlights the uncertainty surrounding the required qualifications and the lack of a clearly defined profile for school nurses, which led to exclusionary discussions among nurses:

“If I had been a paediatric nurse (I’m a family nurse), I would have had a lot of the concepts already solidified. But communication, collaboration, empathy, and social skills—those are things any nurse already has”(N7)

Despite the challenges, the nurses viewed the project as an opportunity for both professional and personal development:

“I think this can move forward, and I’m going to do everything I can to make the [school nurse] role permanent and ensure it’s here to stay. We’ve worked hard and carved out our space”(N2)

Workload

Another prominent issue was the workload and lower financial compensation:

“Sure, I’m closer to home, but I work three times as much. I work morning, afternoon, and night. I even dream about the workshops. I prepare everything at home”(N2)

“I work until three, but I still get calls and have to attend meetings with teachers or headteachers due to issues that arise …”(N1)

Role of School Nurses: Teachers’ perspectives

For teachers, the theme Role of School Nurses focused on two interconnected subthemes: Importance of the Presence of School Nurses and Approach to School Nursing. Teachers emphasised the value of having nurses in schools, providing a sense of security:

“The reassurance and safety they’ve brought to all the teaching staff and management has been incredible”(T1)

“If there’s no healthcare professional on-site, it’s the teachers who have to step in, which is risky due to limited training. Healthcare for minors is a task for specialists”(T6)

“The only downside I see is that they’re not at the school every day”(T8)

Overall, teachers had a limited understanding of the functions, often associating them solely with healthcare tasks:

“There’s a lack of information about what the school nurse’s responsibilities are”(T5)

Despite this, teachers recognised the potential for nurses to contribute to school health education:

“The nurse can support teachers, students, and families in improving health and quality of life through prevention, care, and identifying health issues. It’s a benefit for the entire educational community”(T6)

“Hopefully, in 3 years, having a nurse in school will be normal”(T10)

#### 3.2.2. Theme: School Nursing Project

Nurses and teachers alike highlighted both positive aspects and areas for improvement within the School Nursing Project, addressing topics such as working conditions and the integration of nurses into schools.

Perceptions of Project Implementation

Nurses shared their perceptions of the project’s implementation, which was influenced by institutional characteristics and the challenges posed by insularity:

“Insularity means that, since this is such a huge project run by the Canary Islands Health Service, I completely understand that all the islands need to be involved, but each island has its own particularities, making things complicated”(N7)

The complexity of planning and resource allocation created inconsistencies:

“Each school has its own characteristics. Some don’t have any multiculturalism, while others have 72 different nationalities …”(N5)

Nurses’ experiences also varied depending on school conditions:

“Structurally, the physical space matters. For instance, I have a bathroom and a sink in my office, but no scales or height measurements—many things are missing in other schools. I’m sure next year we’ll have everything, but for now, it’s been a limitation”(N4)

“They’ve given me a huge consultation room, beautifully set up, full of light—everything’s perfect”(N6)

Identified Weaknesses, Improvements Needs, and Suggested Improvements

Nurses identified the lack of time and resources as major weaknesses, hindering proper implementation. The overwhelming workload from training and tasks prevented efficient performance, exacerbated by inadequate school infrastructure such as designated spaces and basic resources:

“There should have been better coordination from the Regional Ministry of Education to ensure spaces were ready for our work. Without a designated classroom, it’s difficult”(N2)

Additionally, the lack of access to relevant student data, such as chronic conditions, created further challenges:

“Some schools still don’t have data on children’s chronic condition. Other nurses elsewhere had the student list from day 1”(N3)

Another major issue was the need to improve integration and belonging within educational institutions:

“It feels like they don’t see the project as their own. It’s a healthcare initiative from the Canary Islands Health Service, and (the Regional Ministry of) Education provided the facilities, but they don’t want to be involved beyond that”(N7)

Time Management and The School Nurse Profile

Time Management and The School Nurse Profile were closely linked, highlighting the importance of coordination and planning for the project’s success. Nurses noted that increased continuity and presence would help them build stronger relationships with students and teachers:

“If I knew I’d have one school next year and could come every day, the children and teachers would get to know me. That seems important”(N2)

They also faced challenges supporting diverse student needs, such as those with autism spectrum disorders:

“We have children with many issues; children with autism, and we’ve had to quickly learn to use pictograms and other tools”(N7)

The absence of a clear structure made it difficult to plan effectively:

“We don’t have an organisational chart because we don’t have a work plan. Hopefully, next year, everything will be better thought out, so the nurse knows exactly what to do”(N2)

Improving Children’s Health and School Health

Teachers, on the other hand, appreciated the project’s positive impact on children’s health and the school environment, particularly the collaboration between health and education:

“I hope we can keep collaborating in the same way and even expand it, so we can have the nurse at the school every day”(T8)

However, teachers also pointed out the need for better coordination and expressed a desire for greater participation:

“We didn’t know exactly what their roles would be, but we were longing for the presence of a school nurse … We also hoped for better coordination between the health and education departments”(T2)

Overall, while the project was seen as beneficial, adjustments in coordination, planning, and resources were suggested to maximise its impact:

“Over the year, we’ve participated in four workshops. We could do more, but with the curricular load and other events, there’s just not enough time”(T4)

## 4. Discussion

Schools are ideal settings for fostering health literacy from an early age, equipping students with the knowledge, motivation, and skills needed to access, understand, and apply health-related information. This empowers children to make informed decisions about health promotion and disease prevention, thereby improving their quality of life [30]. Against this background, adolescence may be too late to acquire and develop healthy habits, further highlighting the value of schools as prime environments for early intervention in health education during early years and primary education [17].

Despite the strategic role of school nurses in health promotion, their integration and utilisation in schools remain suboptimal [31]. While the role of the school nurse has gained significance in recent years, the scope of their responsibilities continues to evolve in line with shifts in educational approaches [32]. However, there is still a need to expand practical experiences and deepen the understanding of the perspectives of nurses, teachers, students, families, and the wider school community [17]. Such efforts are essential for advancing a legislative framework to regulate and strengthen school nursing [32].

In our research, two main thematic areas were identified: Role of School Nurses and School Nursing Project. Regarding the first theme, nurses prioritise employment-related aspects such as the selection process for suitable profiles and professional opportunities. School nursing combines specific skills and roles from other specialisations or fields, such as paediatric nursing, family and community nursing, emergency care, mental health, and public health [32], making the selection of appropriate profiles a complex task. As such, it requires broad and diverse training, with its distinctive and specific feature being the focus on student health within an educational setting [32]. In this regard, mentorship among school nurses is a valuable strategy to increase confidence in practice and develop professional relationships [33], achieved through close collaboration between different profiles and areas of specialization [34]. However, none of the school nurses included in the program were specialists in paediatric nursing, which contrasts with the findings from the interviews, as the participants themselves emphasized the importance of having specific competencies in the field of paediatrics. This aspect should be considered when establishing selection criteria for professional profiles. Furthermore, Armas et al. [34] highlight the importance of recognizing school nursing as a specialized field that needs to be integrated and regulated within healthcare policies and legal frameworks. They propose the establishment of formal mandates to govern the practice of school nursing and validate the unique skills and responsibilities of school nurses. According to these authors, policymakers and relevant authorities should consider implementing legal frameworks that clearly define the scope of school nursing practice, ensuring clarity regarding its functions and responsibilities. Additionally, the importance of school nurses possessing leadership skills is emphasised [31]. These skills are critical for fostering effective collaboration within the educational community, supporting students’ growth and development through individual guidance, and promoting effective coping strategies in the school environment. This approach requires a holistic perspective and an open, cordial atmosphere based on reciprocity and respect for students’ privacy [35].

On the other hand, teachers expressed greater concern about the importance of school nurses’ presence and perspectives within schools, advocating for a child-centred approach that extends beyond addressing immediate health needs. Achieving this requires increasing the number of school nurses and expanding their competencies [31], moving beyond the assessment of healthcare needs to meet the demands of policymakers [36]. Additionally, nurses must feel comfortable and confident in these work environments [36], which necessitates structural and organisational changes within schools. These include improvements in communication and coordination among the various stakeholders involved [31]. It remains essential to develop a specific paradigmatic framework to propose general theoretical models that can guide the practice of school nurses [37].

Regarding the School Nursing Project, nurses identified weaknesses and improvement needs, particularly in time management and resource allocation. The literature highlights a general lack of robust research in this area, underscoring the need to implement quality standards to effectively evaluate the impact of school nurses [38]. For instance, the cost-effectiveness of school nurses has been found to be limited in managing students with chronic conditions and reducing school absenteeism [39]. According to Rankine et al., insufficient communication among professionals is the primary barrier to addressing absenteeism [40]. Leach et al. [39] suggest this issue stems from an ongoing paradigm shift in the care provided by school nurses moving into environments that are markedly different from those shaped by traditional approaches to care and teaching. Chabot et al. argue that increasing the number of school nurses could enhance their sense of value and, consequently, their job satisfaction in the school setting [31].

Teachers, on the other hand, focused on more pragmatic aspects of the project, directing their comments towards their expectations and experiences with the pilot programme and emphasising the importance of interprofessional collaboration. Lam et al. [41] highlight the advantages of such collaboration, including the early identification of children’s complex health needs, promoting health, and ensuring rapid referral when necessary. According to Lam et al. [41], this requires comprehensive planning and implementation of an increased number of school health services. Through these collaborative experiences, policymakers can better understand how interprofessional collaboration within the educational setting can promote child health, underscoring the need for regional and national policy development [41]. For Herath et al. [42], interprofessional collaboration in education represents an innovative strategy that plays a significant role in addressing global shortages of healthcare workers and professionals. However, its implementation varies significantly across countries, with slower progress in developing nations and in rural areas of developed countries [43,44]. Moreover, health programmes are currently more prevalent in higher education settings, making it necessary to strengthen their presence in early years and primary education. According to Herath et al. [42], nursing is the discipline most actively involved in these programmes on an international scale.

The findings of this research are expected to provide a foundation for identifying potential weaknesses, needs, and areas for improvement in developing the profile of school nurses in the Canary Islands. This will help establish strategies to efficiently address the health needs of the school population, the educational community, and the wider public. These strategies should aim to promote healthy habits, foster self-management of chronic diseases, and provide appropriate assistance in emergency situations within educational centres [41]. However, these interventions must be carefully designed and implemented to ensure meaningful outcomes.

Maughan and Baltag [45] identify three primary barriers to the development of comprehensive school nurse profiles: traditional perceptions and a limited understanding of the ways school nurses address contemporary challenges, insufficient systemic support for their integration into educational frameworks, and a lack of robust data demonstrating their value in schools. This research seeks to contribute to the body of knowledge regarding the role of school nurses within the international context, extending its scope beyond the Canary Islands. The findings of the study highlight the role of school nurses in improving student health outcomes and emphasize the need for strategic policy interventions that ensure their integration into the educational system [46]. Based on these findings, it is recommended that policies be developed to allocate adequate resources, provide specialized training, and foster stronger collaboration between the health and education sectors. Practical interventions should focus on increasing the presence of school nurses, enhancing their professional profiles, and involving educators in health planning activities to ensure a more holistic approach to student well-being.

Limitations of this research include the temporary nature of the project and the discontinuous presence of nurses in individual schools, as they are shared across multiple educational institutions simultaneously. This situation necessitates prioritising schools with greater health and social care challenges to achieve better cost–benefit ratios. Other limitations stem from the exploratory nature of the research, characteristic of a qualitative methodology, which does not provide generalisable evidence. Additionally, transferability is influenced by the sociocultural uniqueness of the educational processes studied. Despite these limitations, we believe that our study offers valuable clinical implications by identifying analytical categories that support the development, implementation, and improvement of new health education strategies through the enhanced competencies of school nurses in early years and primary school settings.

## 5. Conclusions

The findings highlight the positive perceptions of nurses and teachers regarding the implementation of the school nursing pilot programme in primary education in the Canary Islands, underscoring its transformative impact on the promotion of health and student well-being. The results indicate that the programme has made significant progress in integrating nurses into the educational system, although key areas for improving the programme’s sustainability and effectiveness have been identified.

The findings reflect that school nurses play a critical role, not only in clinical care but also in health prevention and promotion. However, to enhance their impact, it is necessary to improve working conditions, including workload management, adequate resource allocation, and a clear organizational structure. Effective integration requires specialized training in areas such as paediatrics, health education, and cultural diversity. These improvements would not only ensure the programme’s sustainability but also lead to a more significant impact on school health.

The programme’s implementation highlights the importance of inter-professional collaboration between the healthcare and education sectors. To optimize this synergy, it is essential to strengthen institutional coordination and encourage the participation of teachers in the joint planning and execution of health promotion activities.

Furthermore, it is recommended to expand the programme’s coverage, ensuring a greater presence and continuity of school nurses in educational centers. This would help consolidate the confidence of the educational community, improve the management of chronic health issues, and foster healthy habits from an early age.

In summary, this pilot programme represents a significant advancement and illustrates the potential of nurses to transform school environments into healthy spaces conducive to comprehensive health promotion. Its long-term success will depend on the implementation of strategies that strengthen the nurses’ professional profiles, improve inter-institutional coordination, and ensure the provision of adequate resources.

## Figures and Tables

**Table 1 nursrep-15-00048-t001:** Semi-structured interview script for nurses and teachers.

Participants	Questions
Nurses	Can you describe your experiences as a participant in the school nursing pilot programme?
	What challenges or weaknesses have you identified in the implementation of the school nursing pilot programme?
	What improvements are needed to further develop the role and profile of school nurses?
	Are there specific areas within the pilot programme require enhancement to ensure its success?
Teachers	How would you describe your experiences with the school nursing pilot programme at your school?
	What shortcomings have you noticed in the programme’s implementation, and what areas do you think could be improved?
	What do you see as the main needs for advancing the integration of school nurses into the educational system?
	How do you perceive the role of school nurses, and what impact do you think the programme has had on the school community?
	Which aspects of the programme have been most effective, and what suggestions would you make to strengthen the school nurse role?

**Table 2 nursrep-15-00048-t002:** Sociodemographic characteristics of participating nurses.

Nurse	Island	Sex	Years of Experience	Main Primary Care Facility	Postgraduate Education	Specialization in Nursing	Assigned Schools
N1	La Gomera	Female	3	San Sebastián de la Gomera	Yes	No	2
N2	La Palma	Female	31	Tijarafe	Yes	Family and Community	2
N3	El Hierro	Female	19	Valverde	Yes	Family and Community	4
N4	Tenerife	Female	26	Icod	Yes	No	4
N5	Lanzarote	Female	34	Costa Teguise	Yes	Mental Health; Family and Community	3
N6	Fuerteventura	Female	32	Península de Jandía	No	Family and Community	4
N7	Gran Canaria	Female	21	San Gregorio	Yes	Family and Community	2

**Table 3 nursrep-15-00048-t003:** Sociodemographic characteristics of participating teachers.

Teacher	Island	Sex	Years of Experience	Class Tutor	Teaching Level and Subjects Taught	Number of Students
T1	Fuerteventura	Female	33	No	Primary education	402
T2	Fuerteventura	Female	5	No	Primary education, SEN	402/284 *
T3	Gran Canaria	Female	22	No	Primary education, SEN, Music	402
T4	Gran Canaria	Female	9	Yes	Primary education, SEN, English	402
T5	Gran Canaria	Male	8	No	Primary education, SEN, PE	506
T6	Tenerife	Female	16	No	Primary education	420
T7	Fuerteventura	Female	4	No	Primary education, Music	210
T8	Fuerteventura	Male	16	No	Primary education, English	210
T9	Gran Canaria	Female	25	No	Primary education, SEN, Natural Sciences	411
T10	Tenerife	Male	7	No	Primary education, PE, Mathematics	51
T11	La Gomera	Female	18	No	Primary education, SEN	455
T12	La Palma	Female	11	No	Primary education, SEN	241
T13	La Gomera	Female	1	Yes	Primary education, SEN	455
T14	El Hierro	Male	37	No	Primary education, Mathematics	96

Note: * = Teaches in two schools. SEN = Special Educational Needs. PE = Physical Education.

**Table 4 nursrep-15-00048-t004:** Themes, subthemes, and units of meaning.

Themes	Participants	Subthemes	Units of Meaning
Role of School Nurses	Nurses	Workload	Busy workload, Clear work plan, Need for organisation, Work plan, Prepared workshops, Stable schedule, Variable expectations
		School nurses’ experiences	Career advancement, Career development, Career opportunities, Experiences, Feeling excluded, Increased workload, Integration, Isolation, Motivations, New opportunities, Noise, Occupation, Positive experiences, Professional growth
		Career opportunities	Accidental discovery, Addressing expectations, Career advancement, Career advancement opportunities, Career development opportunities, Career opportunity, Changing expectations, Considering to participate, Deciding to participate, Education opportunities, Clarification of expectations, Feeling isolated, Future career development, Social mobility opportunities, Motivations for participation, Need for improvement, New opportunities, Opportunity for career advancement, Opportunity for career growth, Positive experiences, Programme benefits, Training, Variable experiences
		The school nurse profile	Addressing needs, Clarifying expectations, Changing expectations, Collection of relevant data, Defining clear expectations, Developing the profile of the school nurse, Rewarding, Improved training, Improving the profile, Inter-professional coordination, Lack of clear expectations, Motivational workshops, Motivators, Pilot programme, Positive experiences, Profile development needs, School nurse training, Strengthening the profile of the school nurse, School nurse, Standardisation of expectations, Varied expectations
	Teachers	Approach to school nursing	Clinical and pedagogical approach, Clinical aspect, Clinical point of view, Necessary aspect, Pedagogical aspect, Pedagogical point of view
		Importance of the presence of school nurses	Collaboration, Education, Importance, Positive, Potential, Presence, Relevance, Healthcare worker
School Nursing Project	Nurses	Suggested improvements	Effective communication, Institutional support, Motivation, Organisation, Planning, Recognition, Selection process, Task sharing, Training, Time management, Work plan, Working time, Workload management
		Identified weaknesses	Lack of support, Lack of resources, Lack of time, Motivation, Sense of belonging, Work overload
		Time management	Allocation of adequate study time, Attention to students, Busy, Excessive workload, Implementation of workshops, Increased workload, Intensive training load, Lack of integration, Lack of time to study, Organisation of study time, Programme organisation, Resource allocation, Recognition of work, Tasks required, Time management, Training load, Time to study, Time to study at home, Workload management, Workshop preparation
		Improvement needs	Better organisation, Clear expectations, Continuous training, Effective communication, Integration, Updating knowledge, Planning, Time management, Working hours, Work overload, Workshops, Work plan, Workload management
	Teachers	Expectations	Expectations
		Improving children’s health	Rewarding, Health care, Health and education, Positive collaboration, Positive experiences
		School health	Educational institutions, Educational resource, Pedagogical needs, School health, School nurses
		Experiences from the pilot programme	Changes, Collaboration, Education, Educational need, Health, Positive experience

## Data Availability

The data used in this research are confidential and stored in a coded and anonymized database maintained by the research group in compliance with Spanish regulations. However, raw data, including interview transcriptions, units of meaning, and thematic areas, can be shared with researchers who contact the corresponding author, provided they offer a reasoned and justified request.

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
