# Peer review of "Perceptions on the Implementation of a School Nursing Pilot Programme in the Canary Islands"

_nursrep, 2025, doi:10.3390/nursrep15020048_

Round 1
Reviewer 1 Report
Comments and Suggestions for Authors
Thank you for the opportunity to review this article.
The manuscript addresses a theme of extreme relevance for nursing and health promotion of schoolchildren.
The introduction/outline of the object of study and the objective of the article were clearly presented. I only suggest revisiting some references used in paragraphs 1 and 2 of the introduction (citations 3 to 14). Some of these citations could be suppressed and/or replaced by more recent and available literature on the subject.
The method could be improved in the description of the following items:
1. Study scenario: describe in more detail the characteristics of the Canary Islands and the distribution of schools, population profile (natives, immigrants), cultural customs, among others.
2. Sampling: The authors mention that the selection of participants occurred initially by convenience, followed by theoretical sampling and saturation. But what criteria defined the choice of the first participants interviewed? What logic was used by the researchers for this recruitment, since there was saturation and not all the population (which is small) needed to be recruited?
3. Data collection: was there a recording of the face-to-face and virtual interviews? What kind of recording, if the answer is yes (audio? Video? Both?). If the recording has occurred, how is the transcription performed? Was there any software used for tsnscription? Were any other collection techniques used by the researchers to triangulate the data as a field diary or other resource? In line 138, the authors mention that they performed the triangulation of researchers and data to interpret the data.
Discussion: The authors collected information about nurses related to professional experience and post-graduation, but did not mention which specialties the nurses had. This data may be an important factor in determining how the professional experience of school nurses in the government project took place. Another important point is that the authors themselves highlight some areas and disciplines that can contribute to the development of school nursing competencies, but there is no dialogue between the data related to the sample profile and what the literature brings as evidence. Thus, the data presented in the characterization is loose and did not contribute to the interpretation/discussion of the research results.
Reviewer 2 Report
Comments and Suggestions for Authors
This manuscript presents a relevant qualitative study exploring the experiences of nurses and teachers in a school nursing pilot program. The topic is significant for advancing health promotion in educational settings and contributes valuable insights for policymakers and healthcare professionals.
Here there are suggestions to improve the manuscript: Introduction: The introduction provides a solid background on the role of school nurses; however, a clearer explanation of the research gap that the study seeks to fill would be beneficial. Consider elaborating on why the Canary Islands context is unique and warrants investigation. Ensure all key references are cited appropriately to support the claims made about the role and impact of school nursing. Research Design: The phenomenological approach is well-justified; however, providing additional details on why van Manen’s approach was selected would strengthen the methodological transparency. Please, clarify whether participants were aware of the study's aim before the interviews and how potential bias was managed. Methods: The methods section describes the data collection process adequately but it is better if the authors added information about the interviewers' training and their potential influence on the data collection process. Please, add more detail on the coding process and how reliability is ensured would enhance methodological rigor. Results: The results section is comprehensive, with clear themes and subthemes identified. However, the clarity of some quotes could be improved by reducing redundancy and emphasizing key findings more directly. Consider rephrasing the co-occurrence data for better clarity, ensuring that it is accessible to readers unfamiliar with qualitative analysis software. Discussion: The discussion effectively relates the findings to the existing literature but could further explore implications for international contexts beyond the Canary Islands. Please, clarify how the findings contribute to policy recommendations and practical interventions for integrating school nurses more effectively. Conclusions: The conclusions align with the study findings but should more explicitly address the identified limitations and their impact on the study's generalizability.
Round 2
Reviewer 1 Report
Comments and Suggestions for Authors
I would like to thank the authors for elucidating my doubts about some topics in the manuscript and I am happy to contribute by realizing that the adequacy of the wording and elucidation of some gaps observed throughout the text. The article has become clearer and better conveys the essence of the research, especially in the stages of method and discussion/conclusion.